# The gender gap in aversion to COVID-19 exposure: Evidence from professional tennis

Zuzanna Kowalik[1☯], Piotr Lewandowski[1,2☯] *

**1** Institute for Structural Research (IBS), Warsaw, Poland, **2** IZA, Bonn, Germany

☯ These authors contributed equally to this work.
* piotr.lewandowski@ibs.org.pl

**Data Availability Statement:** All relevant data are within the paper and its Supporting Information files.

**Funding:** The authors received no specific funding for this work.

## Abstract

We study the gender differences in aversion to COVID-19 exposure using a natural experiment of the 2020 US Open. It was the first major tennis tournament after the season had been paused for six months, held with the same rules and prize money for men and women. We analyze the gender gap in the propensity to voluntarily withdraw because of COVID-19 concerns among players who were eligible and fit to play. We find that female players were significantly more likely than male players to have withdrawn from the 2020 US Open. While players from countries characterized by relatively high levels of trust and patience and relatively low levels of risk-taking were more likely to have withdrawn than their counterparts from other countries, female players exhibited significantly higher levels of aversion to pandemic exposure than male players even after cross-country differences in preferences are accounted for. About 15% of the probability of withdrawing that is explained by our model can be attributed to gender.

## Introduction and motivation

The COVID-19 pandemic has affected men and women differently. Compared to men, women are less likely to become severely ill or die from COVID-19 [1], but are as likely to be infected. In 35 high-income countries with available data, women constitute the majority (52%) of those infected (according to the COVID-19 Sex-Disaggregated Data Tracker data). Gender differences in engaging in workplace interactions that are critical for the spread of infectious diseases transmitted by the respiratory or close-contact route, such as COVID-19 [2], are among the social factors that contribute to this gender gap. Compared to male workers, female workers have higher occupational exposure to contagion, and are more likely to work in highly exposed occupations, such as health professionals, care, and personal care workers [3]. At the same time, the COVID-19 crisis has affected the employment outcomes of women more than those of men. This is in part because the crisis hit the female-dominated sectors particularly hard, and, within these industries, women have been more likely than men to lose their jobs [4].

Policy responses from all over the world aim to reduce social contacts and limit contagion, but the compliance with non-pharmaceutical interventions can vary across different groups. Insights from social and behavioural sciences can help to understand the gap between the

**Competing interests:** The authors have declared that no competing interests exist.

recommendations of experts and actual human behavior during the COVID-19 pandemic [5]. Gender is perceived as one of the factors associated with this compliance. Previous research has found that women are more likely than men to perceive the COVID-19 pandemic as a very serious health problem [6] and to engage in preventive behaviours [7]. Women more often follow the public health recommendations, such as wearing face masks [8], especially when these are not compulsory [9]. Also during the past respiratory epidemics, women were found to be 50% more likely than men to adopt non-pharmaceutical protective behaviours [10].

Overall, women appear to be more concerned with the pandemic. An important question that arises in this context is whether these potential gender differences in perceptions of COVID-19 risk affect the labor market participation of men and women.

In this paper, we contribute to the literature on the gender dimension of the COVID-19 pandemic by studying the gender gap in aversion to pandemic exposure. We use a natural experiment of the US Open tennis tournament held in New York City between August 31 and September 13, 2020. It was the first major tournament that was organized after the tennis season had been put on hiatus due to COVID-19 concerns. We analyze the factors associated with the voluntary withdrawals of players who were eligible to play. The Grand Slam tennis tournaments, like the US Open, constitute a useful setting for studying gender differences in decision-making and performance, as the conditions for participation, the structure of the tournaments, and the prize money amounts are identical for men and women. In 2020, the health and safety protocols at the US Open were also the same for both genders. Hence, the gender differences in the propensity to voluntarily withdraw from the US Open can be interpreted as having been driven by gender differences in aversion to pandemic exposure, especially given that the tournament took place in the country that had the highest numbers of COVID-19 cases and deaths.

We find that female players were significantly more likely to have voluntarily withdrawn from the tournament, which was organized in an epidemiologically risky setting. Higher-ranked players and older players were also more likely to have withdrawn. Our findings are robust to controlling for cross-country differences in cultural preferences. While players from countries characterized by higher levels of patience and trust and by lower levels of risk-taking were also significantly more likely to have withdrawn, the gender gap remains significant even after we control for these factors. It is also found to be particularly large among players ranked in the top 50, who are richer than lower ranked players.

Gender differences in risk-taking have been cited as being among the main causes of gender gaps in employment outcomes, including the gender pay gap and the underrepresentation of women in top-tier jobs [11] Experimental studies have suggested that females are more averse to risk and tend to shy away from competitive settings [12], although the estimated sizes of the gender differences in risk attitudes vary depending on the method used and the context in which decisions are made [13] The real-life studies tend to be focused on gender differences in investment choices in financial markets or pension funds [14]. However, grasping the differences in risk-taking behavior is harder in real-life situations than in lab experiments because outside of the laboratory, the available options are usually not restricted to a well-defined set of choices. Professional sports create opportunities to tackle this challenge [15]. Sports involve rules that aim to ensure that the players have symmetrical information, and to define a set of options that can often be ranked by their level of risk. Incentive structures and rewards are linked to performance, which can be precisely measured. Individual sports such as tennis, are particularly suitable to study strategic behavior [16, 17]. Psychological factors, such as risk preferences, play an important role [18, 19]. We contribute to this strand of literature by studying gender differences in reactions to a specific type of risk related to exposure to a pandemic.

While we do not study the actual performance of players in the tournament, we use its identical treatment of men and women to assess gender differences in the willingness to participate in a professional activity perceived as risky and inconvenient due the COVID-19 pandemic.

## Data and methodology

The professional tennis season was paused on March 12, 2020, due to concerns about COVID-19. The female tour resumed on August 3, 2020, with tournaments in Palermo and Prague, followed by the Western & Southern Open and the US Open tournaments played at the same venue in New York City. The male tour resumed on September 22, 2020, with the Western & Southern Open tournament, followed by the US Open, which were played at the same venue, and parallel to the female tournaments. During the tournaments in NYC, a "bubble" was created and the same health and safety rules were applied to all male and female players: (i) each player's entourage could include up to three people, and (ii) the players had to stay in one of the two pre-approved hotels. The players were not allowed to leave the bubble, or to have contact with any people outside of it. Audiences were not allowed in the stadiums.

The Grand Slam tournaments such as the US Open have an identical structure for male and female competitions: 128 players are included in each draw, and the prize money is the same for men and women. In general, players ranked in the top 100 before a Grand Slam tournament are automatically eligible to play, while the remaining places are awarded to players who are successful in a qualifying tournament, and players who receive the so-called wildcards. In 2020, the qualifying tournament was not held, and entry lists were created based on rankings. However, we restricted our study to the top 100 male and female players, as these players could have expected long in advance that they would be eligible to play. To collect information about rankings, we scraped data from websites of Association of Tennis Professionals (www.atptour.com) and Women Tennis Association (www.wtatennis.com), and we merged it with the entry list to the US Open (www.usopen.org).

In 2020, 36 out of the top 100 male and female players did not participate in the US Open. Of these players, 25 (19 women and six men) withdrew due to COVID-19 concerns (Fig 1). Another 11 players withdrew for other reasons. For each player, we specified the reason for his/her withdrawal using his/her social media posts and/or media interviews (see S1 and S2 Tables, sources available upon request). Our final sample includes 189 players from 48 countries who were eligible and fit to play.

In order to analyze the differences between female and male players in the propensity to withdraw from performing in a pandemic environment, we estimated logistic (1) regressions:

$$\Pr\left(\text{withdraw from US Open}_{jc} = 1\right) = F(\beta_0 + \beta_1 X_j + \beta_2 \lambda_c + \varepsilon_{jc}) \tag{1}$$

where $F(Z) = \frac{e^Z}{1+e^Z}$, $j$ stands for individual, and $c$ for country; $X_j$ is a vector of personal characteristics (sex, age (log), ranking (log)); and $\lambda_c$ is a vector of country-level controls.

As noted by Van Bavel et al. [5], some differences in the response to the pandemic may be described as cultural. Bargain and Aminjonov [20] showed that European regions with higher levels of political trust recorded significantly larger mobility reductions, and more pronounced effects of the containment policy stringency, so trust appears to be related to compliance. In order to account for cultural factors, we controlled for country-level measures of patience, risk-taking, altruism, and trust based on the Global Preference Survey conducted in 76 countries [21] to account for the general willingness to take risks, as well as in other preferences observed differences in preferences that may affect attitudes toward the pandemic and compliance with containment policies. The GPS data are not available for the following countries that include a total of 21 top 100 male or female players who were eligible and fit to participate:

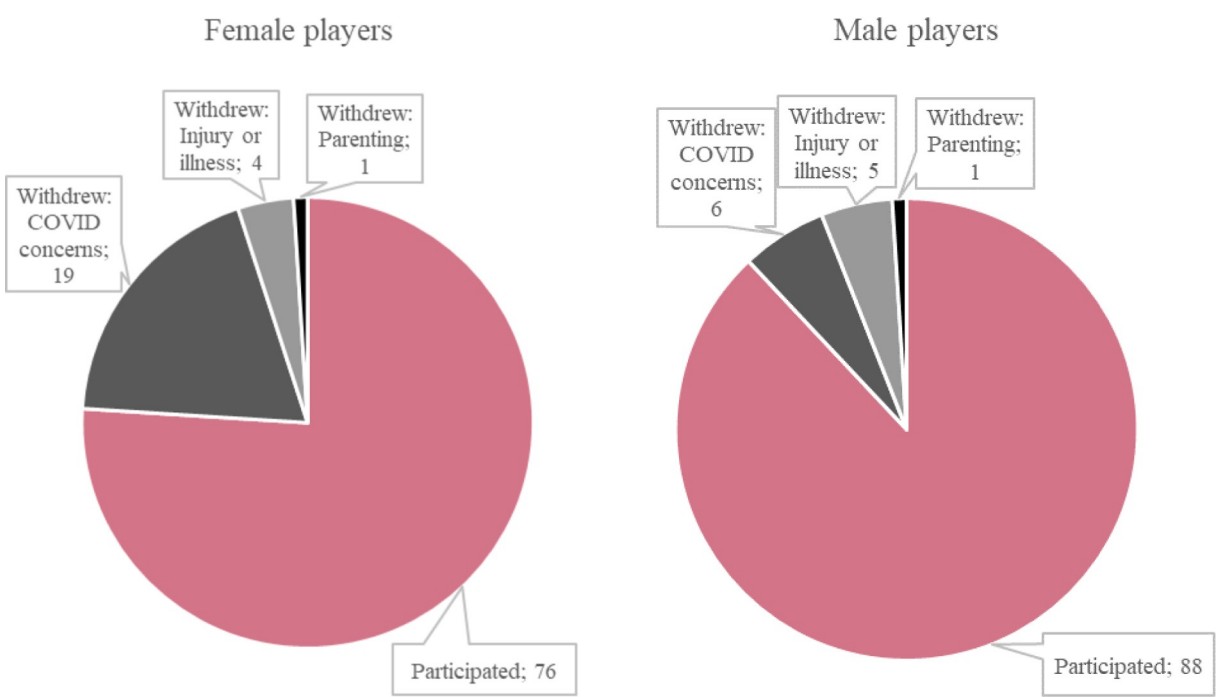

**Fig 1. Structure of top 100 players by the US Open status.** Source: Own elaboration based on webscraped data.

Belgium, Belarus, Latvia, Montenegro, Norway, Puerto Rico, Slovenia, Slovakia, Taipei, Tunisia, and Uruguay. We also controlled for GDP per capita (in purchasing power parity, log, from the World Development Indicators database), as gender differences in preferences may be expressed more frequently in more developed than in less developed countries [22]. We standardized all country-level variables, ages, and rankings.

We estimated two variants of model (1). First, we controlled for individual characteristics only. Second, we added country-level controls. We also re-estimated our model on a subsample of non-US players, for whom the perceived risk of participation may be higher because they have to undertake international travel and may be unfamiliar with the local health system; as well as for subsamples of players ranked in the top 50 and in places 51–100. In order to test whether our findings are robust to changes in estimation method, we also estimated probit and linear probability models instead of a logit model.

Finally, in order to assess the relative role of gender and other factors in the probability of withdrawing from the 2020 US Open, we used the Shapley decomposition proposed by Shorrocks [23]. For each factor, the marginal impact on the withdrawal probability is calculated as the factors are eliminated in succession. Next, these marginal effects are averaged over all the possible elimination sequences. The contributions sum to the total probability explained by a model, and they can be interpreted as expected marginal effects. This method originated with poverty decompositions, but can be applied to any econometric specification [23].

## Results

We begin with descriptive evidence. In 2020, the total number of players who withdrew from the US Open (36) was more than double the number in 2019 (17). In 2019, 17 out of the top 100 male and female players withdrew from the US Open because of injuries or for personal reasons. In 2020, the number of injured players was lower than it was in 2019, as there was no

**Table 1. Descriptive statistics.**

|  | Female | Age | Ranking | Patience | Risk taking | Altruism | Trust | GDP per capita |
|---|---|---|---|---|---|---|---|---|
| Participant | 46.3% | 26.9 | 53 | -0.02 | 0.03 | 0.00 | -0.04 | 0.00 |
| Withdrawn | 76.0% | 29.1 | 34 | 0.21 | -0.04 | 0.11 | 0.45 | -0.16 |

Note: All country-level variables- patience, risk taking, altruism, trust and GDP per capita are standardized in our sample.

Source: Own calculations based on data web scraped from US Open, ATP and WTA websites, GPS, and WDI data.

competition between March and August which reduced the risk of injury. Thus, the increase in the total number of withdrawals was driven by voluntary withdrawals which suggests that the pandemic increased the propensity of players to avoid performing.

The share of women among the top 100 players who voluntarily withdrew from the 2020 US Open was noticeably higher (76%) than it was among the players who participated (46%, Table 1). Moreover, the players who withdrew were, on average, older and higher ranked than the players who participated. These players also came disproportionately from countries that exhibit higher levels of patience, lower levels of risk-taking, higher levels of altruism and trust, and lower levels of GDP per capita.

Next, we present our econometric results. We find that female players were significantly more likely (by 15.3 percentage points on average) to decide against participating in the US Open than men (column 1 of Table 2). Older players, and higher ranked players were also more likely to withdraw. The probable mechanism behind the effect associated with ranking is that higher ranked players are more affluent and earn higher incomes from endorsements, and thus may have been more willing to forego prize money from a tournament perceived as risky. The size of the effect associated with gender is found to be slightly smaller (12.3 pp) in a sub-sample of players for which the Global Preference Survey data are available (column 2 of Table 2). While controlling for country-level covariates reduces the size of the effects associated with gender, it remains highly significant and large (10.2 pp, column 3 of Table 2). Moreover, players from countries characterized by higher levels of patience and by lower levels of risk-taking were significantly more likely to have decided against participating in the 2020 US Open (column 3 of Table 2). This means that part of the differences in the rate of withdrawals among female and male players can be attributed to the fact that women players happen to more often come from countries characterized by stronger preferences conducive to avoiding exposure, such as higher patience and lower risk aversion. However, female players were significantly more likely to withdraw even if these differences in country-level preferences are factored in.

Importantly, in a subsample of non-US players, the effect associated with gender is shown to be noticeably larger (15.1 pp) than in the full sample (column 4 of Table 2). At the same time, the effects associated with other factors at both the individual and the country level are found to be similar to those estimated in the full sample. This suggests that the female players may have been particularly more concerned about exposure associated with travel than the male players.

Finally, the female players were significantly more likely to have voluntarily withdrawn than the male players among the top 50 players (columns 5 and 6 of Table 2), but among the players ranked 51–100 the effect is not significant (columns 7 and 8 of Table 2, respectively). This difference in the effect associated with gender is likely to be related to the income and wealth discrepancies between higher and lower ranked players. Professional tennis exhibits rather large concentration of incomes among those at the top. The top 50 players (who constitute 1% of all players) earn more than 50% of all prize money, and the average earnings of top 50 players are about five times the average earnings of players ranked 51–100, both among

**Table 2. The correlates of voluntary withdrawal from the 2020 US Open due to COVID-19 concerns (marginal effects).**

| | Player controls | Player controls (sample w/ GPS data) | Player and country controls | Player and country controls, no US players | Player controls, players ranked in top 50 | Player and country controls, players ranked in top 50 | Player controls, players ranked 51–100 | Player and country controls, players ranked 51–100 | Player and country controls (probit) | Player and country controls (OLS) |
|---|---|---|---|---|---|---|---|---|---|---|
| | (1) | (2) | (3) | (4) | (5) | (6) | (7) | (8) | (9) | (10) |
| Female | 0.153*** | 0.123** | 0.102** | 0.151*** | 0.180** | 0.148* | 0.064 | 0.054* | 0.092** | 0.101** |
| | (0.051) | (0.050) | (0.048) | (0.044) | (0.079) | (0.079) | (0.053) | (0.030) | (0.044) | (0.046) |
| Age (log) | 0.069** | 0.071** | 0.067** | 0.101*** | 0.110** | 0.101** | 0.035 | 0.044 | 0.068*** | 0.070** |
| | (0.028) | (0.029) | (0.030) | (0.032) | (0.048) | (0.050) | (0.029) | (0.031) | (0.026) | (0.027) |
| Ranking (log) | -0.067*** | -0.069*** | -0.068*** | -0.082*** | -0.075** | -0.076** | -0.012 | -0.018 | -0.071*** | -0.103*** |
| | (0.018) | (0.018) | (0.016) | (0.016) | (0.033) | (0.031) | (0.102) | (0.102) | (0.017) | (0.031) |
| Patience | | | 0.092** | 0.101*** | | 0.109* | | 0.070 | 0.092*** | 0.089** |
| | | | (0.037) | (0.033) | | (0.062) | | (0.055) | (0.035) | (0.035) |
| Risk-taking | | | -0.065** | -0.049* | | -0.095* | | -0.048 | -0.059** | -0.042** |
| | | | (0.031) | (0.028) | | (0.057) | | (0.034) | (0.029) | (0.021) |
| Altruism | | | -0.005 | 0.073** | | -0.007 | | 0.000 | -0.007 | -0.000 |
| | | | (0.023) | (0.034) | | (0.044) | | (0.014) | (0.023) | (0.022) |
| Trust | | | 0.042* | 0.008 | | 0.046 | | 0.046** | 0.043** | 0.052* |
| | | | (0.022) | (0.028) | | (0.046) | | (0.020) | (0.022) | (0.027) |
| GDP per capita (PPP, log) | | | -0.061*** | -0.035 | | -0.054 | | -0.043** | -0.061*** | -0.090** |
| | | | (0.023) | (0.026) | | (0.054) | | (0.022) | (0.023) | (0.042) |
| Adj. R2 / R2 | 0.185 | 0.215 | 0.308 | 0.412 | 0.189 | 0.241 | 0.103 | 0.472 | 0.315 | 0.233 |
| No. of obs. | 189 | 168 | 168 | 142 | 86 | 86 | 82 | 82 | 168 | 168 |

Note: All parameters are presented as average marginal effects. All models include constant (not presented). R2 shown in the case of linear probability model (8). Robust standard errors in parentheses.

*** p<0.01

** p<0.05

* p<0.1.

Source: Own estimation based on data web scraped from US Open, ATP and WTA websites, GPS, and WDI data

women and men [24]. Hence, the top 50 players may be more prepared to forego income from a given tournament perceived as risky. We find that the gender gap in aversion to pandemic exposure is manifested to a larger extent among players with higher ranking and incomes. It is consistent with Falk and Hermle [22] who show that gender differences in preferences are more likely to be revealed when incomes are higher.

Our findings are robust to changes in the econometric methodology, as the probit and linear probability models (columns 9 and 10 of Table 2, respectively) deliver similar estimates as the logit model (column 3 of Table 2).

Having established the statistical significance of particular variables, we move to the decomposition analysis which allows us to assess the magnitudes of various factors in explaining the probability of withdrawal from the 2020 US Open. To this aim, we use the Shapley decomposition proposed by Shorrocks [23]. Overall, our models are able to explain 20–30% of the overall variance of withdrawal probability in different samples (Table 3). It's a relatively high share considering that the decision about participation may be affected by individual factors and past experiences [25], but our data allow controlling only for sex, age, and ranking, and country-level variables.

**Table 3. The Shapley decomposition of the probability of voluntary withdrawal from the 2020 US Open.**

|  | Gender | Age | Ranking | Preferences (country-level) | GDP per capita | Total |
|---|---|---|---|---|---|---|
| All players |  |  |  |  |  |  |
| Contribution | 0.028 | 0.039 | 0.100 | 0.047 | 0.019 | 0.233 |
| % of explained variance | 12.0% | 16.9% | 42.8% | 20.0% | 8.3% | 100.0% |
| Non-US players |  |  |  |  |  |  |
| Contribution | 0.053 | 0.052 | 0.108 | 0.086 | 0.010 | 0.310 |
| % of explained variance | 17.2% | 16.8% | 35.0% | 27.8% | 3.3% | 100.0% |
| Top 50 players |  |  |  |  |  |  |
| Contribution | 0.039 | 0.066 | 0.061 | 0.047 | 0.008 | 0.221 |
| % of explained variance | 17.8% | 29.7% | 27.6% | 21.2% | 3.8% | 100.0% |
| Players ranked 51–100 |  |  |  |  |  |  |
| Contribution | 0.018 | 0.022 | 0.001 | 0.098 | 0.051 | 0.191 |
| % of explained variance | 9.6% | 11.6% | 0.5% | 51.6% | 26.7% | 100.0% |

Note: Shapley decomposition based on linear probability models. Contribution of preferences is a sum of contributions of patience, risk- taking, altruism, and trust.

Source: Own estimation based on data web scraped from US Open, ATP and WTA websites, GPS, and WDI data.

We find that about 15% of explained variance in the probability of voluntary withdrawal can be associated to gender (Table 3). The contribution of gender is the largest (almost 20%) in the sample of non-US players, and in the sample of top 50 players. It is the smallest in the sample of players ranked 51-100(10%).

Overall, ranking is the factor with the largest contribution (30–40% of explained variance), except for the players ranked 51–100 (0.5%). The contribution of age is particularly high among the top 50 players (30%). Thus, the decision to voluntarily withdraw from a tournament in a country badly affected by the pandemic could have been related to the ability to forego income from this tournament, presumably higher among higher ranked players.

The contribution of country-level cultural preferences is the highest among the players ranked 51–100 (50%)–in this group of players for whom the tournament could have been a key source of income, country-level preferences are by far the most important factor associated with voluntary withdrawals. The contribution of preferences is also high among the non-US players (28%). Our results suggest that differences in preferences may translate into differences in attitudes to the pandemic exposure.

## Discussion and conclusions

In this paper, we have studied the gender differences in aversion to COVID-19 exposure. To do so, we used a natural experiment of the professional tennis US Open tournament, which was the first major tournament organized after the tennis season had paused for six months due to the pandemic. It was held in the country with the highest numbers of COVID-19 cases and deaths, and 14% of eligible and fit players declined to participate. As the conditions and rules for participation, as well as the prize money amounts, were identical for men and women, we have argued that the differences found in the propensity to voluntarily withdraw reflect gender differences in aversion to pandemic exposure.

Our results show that female players were significantly more likely to have withdrawn from the 2020 US Open because of COVID-19 concerns. Players from countries characterized by higher levels of trust and patience and lower levels of risk-taking were more likely to have withdrawn. However, the female players exhibited significantly higher levels of aversion to pandemic exposure than male players, even if cross-country differences in preferences are

accounted for. About 15% of the withdrawal probability explained by our model can be attributed to gender, namely to higher propensity to withdraw among women.

Importantly, the pattern of COVID-19 infections among professional tennis players is in line with our findings: between February and October, 2020, 10 of the top 100 male players, and none of the top 100 female players tested positive. Only one of these players (Benoit Paire) tested positive in the US Open bubble, and none of his contacts have tested positive [26]. This suggests that women players were more likely to engage in preventive behaviors, including withdrawing from a tournament in the US. Previous studies showed that past experiences of negative shocks can increased risk aversion [25] and lower risk taking [27]. However, we think it is unlikely that this experience will affect future risk taking of players, especially of women players who appear more risk-averse, because no outbreaks emerged.

Our study has limitations, in particular related to sample size. However, the US Open is the only natural experiment in tennis that we can study. To our knowledge, no other sport has held parallel competitions with identical rules and prize money amounts for men and women, especially competitions requiring two to three-week long stay at the same venue which amplifies potential pandemic exposure. Moreover, the participation of fit players in the tennis tournaments in Europe, organized after the US Open, was likely affected by the fact that there was no COVID-19 outbreak at the US Open. Indeed, only four players–two Australian (one male, one female) and two Chinese (both female) players–did not participate due to COVID-19 concerns in Roland Garros, the next Grand Slam tournament taking place in Paris on September 27-October, 2020. Finally, our sample size is similar to the one used in past studies of top tennis players [28, 29].

Of course, there may be other hypotheses that can potentially explain gender differences in the propensity to withdraw, career length being one of them. If women had longer professional tennis careers than men, they could perhaps feel they can better accommodate skipping a tournament. In general, women reach their highest career level earlier than men, which is consistent with their more precocious biological development [30]. Indeed, in our sample the average age of top 100 female players (26.9 years) is lower than that of top 100 male players (27.8). Moreover, the average career length of female and male players is not significantly different [30]. Overall, it is unlikely that differences in career life cycle of female and male players can explain different attitudes towards performing during the pandemic.

Our findings have implications for sport competitions as well as for labor markets. Women appear to be more concerned about the pandemic, more willing to comply with preventive measure [6, 7], and more likely to voluntarily avoid performing in a setting perceived as risky. Organizers of sporting competitions should perhaps take these differences into account and adapt the rules and preventive measures to reflect higher level of concern among women. Ignoring these differences may discourage women from performing, which may have negative repercussions for career development, especially among younger players, and may impact negatively on performance if participation is associated with higher level of stress.

Professional tennis players are a relatively well-paid group of people who can afford to avoid participating in tournaments if doing so is perceived as risky, even though because of their young age they face a low risk of severe illness from COVID-19 [1]. In the general labor market, however, women are more exposed to contagion than men because of sectoral and occupational segmentation [3], and can rarely shield themselves from this exposure [4]. However, standard labor market data do not allow studying whether the aversion to pandemic affects willingness to work because most people are not able to freely choose whether they participate in a given work task. Our findings based on a natural experiment suggest that as women have higher levels of aversion to exposure, working women who cannot avoid exposure may experience additional hardships, which may contribute to their reported worse

mental health outcomes during the pandemic [31]. Our results also suggest that focusing on gender differences in labor market outcomes may underestimate the true gender impact of the COVID-19 pandemic in terms of wellbeing. Labour, social and public health policies could account for these gender differences in the design of programs aimed at shielding people from the exposure to infection, as well as mental, social and economic consequences of the pandemic.

## Supporting information

**S1 Table. Players who withdrew from the 2020 US Open because of COVID-19.**
(DOCX)

**S2 Table. Players who withdrew from the 2020 US Open for other reasons.**
(DOCX)

**S1 File.**
(DTA)

## Acknowledgments

We thank Jan Gromadzki for help with data scraping. The usual disclaimers apply. All errors are ours.

## Author Contributions

**Conceptualization:** Piotr Lewandowski.

**Data curation:** Zuzanna Kowalik.

**Formal analysis:** Zuzanna Kowalik, Piotr Lewandowski.

**Methodology:** Piotr Lewandowski.

**Writing – original draft:** Zuzanna Kowalik, Piotr Lewandowski.

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
