## [Decision Letter · Decision Letter 0]

12 Jan 2021

PONE-D-20-38751

The gender gap in aversion to COVID-19 exposure: evidence from professional tennis

PLOS ONE

Dear Dr. Lewandowski,

Thank you for submitting your manuscript to PLOS ONE. After careful consideration, we feel that it has merit but does not fully meet PLOS ONE’s publication criteria as it currently stands. Therefore, we invite you to submit a revised version of the manuscript that addresses the points raised during the review process.

Please find below the reviewers' comments, as well as those of mine.

We look forward to receiving your revised manuscript.

Kind regards,

Valerio Capraro

Academic Editor

PLOS ONE

Additional Editor Comments:

I have now collected two reviews from two experts in the field. The reviewers like the paper but suggest a major revision. Therefore, I would like to invite you to revise your paper following the reviewer's comments. Besides these, I would like to add two more comments, regarding the literature review, which I found to be incomplete. (i) The "perspective article" on what social and behavioural science can do to support pandemic response, published by Van Bavel et al. in Nature Human Behaviour can be a useful introductory reference. (ii) regarding gender differences in pandemic response, I have recently published a paper on gender differences in wearing a face mask and in perceived likelihood to get infected by covid-19, whose results seem to be in line with your results (Capraro & Barcelo, 2020). Of course, it is not a requirement to cite these papers, but I am mentioning them, because they seem very related to yours.

I am looking forward for the revision.

References

Capraro, V., & Barcelo, H. (2020). The effect of messaging and gender on intentions to wear a face covering to slow down COVID-19 transmission. Journal of Behavioral Economics for Policy, 4, Special Issue 2, 45-55.

Van Bavel, J. J., et al. (2020). Using social and behavioural science to support COVID-19 pandemic response. Nature Human Behaviour, 4, 460-471.

Journal Requirements:

3. We note that your paper includes detailed descriptions of individual patients/participants. As per the PLOS ONE policy (http://journals.plos.org/plosone/s/submission-guidelines#loc-human-subjects-research) on papers that include identifying, or potentially identifying, information, the individual(s) or parent(s)/guardian(s) must be informed of the terms of the PLOS open-access (CC-BY) license and provide specific permission for publication of these details under the terms of this license. Please download the Consent Form for Publication in a PLOS Journal (http://journals.plos.org/plosone/s/file?id=8ce6/plos-consent-form-english.pdf). The signed consent form should not be submitted with the manuscript, but should be securely filed in the individual's case notes. Please amend the methods section and ethics statement of the manuscript to explicitly state that the patient/participant has provided consent for publication: “The individual in this manuscript has given written informed consent (as outlined in PLOS consent form) to publish these case details”.

Reviewers' comments:

Reviewer's Responses to Questions

**Comments to the Author**

1. Is the manuscript technically sound, and do the data support the conclusions?

Reviewer #1: Yes

Reviewer #2: Yes

2. Has the statistical analysis been performed appropriately and rigorously? 

Reviewer #1: Yes

Reviewer #2: Yes

3. Have the authors made all data underlying the findings in their manuscript fully available?

Reviewer #1: Yes

Reviewer #2: Yes

4. Is the manuscript presented in an intelligible fashion and written in standard English?

Reviewer #1: Yes

Reviewer #2: Yes

5. Review Comments to the Author

Reviewer #1: General Comments: The current study uses analysis of participation in the 2020 US Open tennis tournament to investigate gender differences in aversion to COVID-19 exposure. The authors are applauded for their novel approach in using the tournament as a natural experiment and the results align with previous literature indicating that women are more likely to engage in protective behaviors (e.g. withdrawing from the tournament). While this work has interesting results, I challenge the authors to consider the implications of their findings more rigorously. Assuming that aversion to exposure to COVID-19 is still relevant as we are still in the throws of the pandemic, how might these results relate to sporting competition at lower levels? Is gender differences in aversion to exposure something that needs to be addressed to facilitate increased participation among females? The authors note that tournaments after the US Open did not see a similar number of withdrawals but this may still be an issue at lower levels of sporting competition. Additionally, the overall flow of the manuscript can be significantly improved to better facilitate readability and comprehension of the material. As noted in my comments, there are multiple paragraphs that seem out of place. Even so, with the necessary revisions I believe this manuscript will make a valuable addition to the COVID-19 literature as it continues to underscore gender differences in the approach to managing the pandemic.

Major Concerns:

• Lines 50-56: As this paragraph involves discussion of the results, it seems it should be moved out of the introduction of the manuscript

• Lines 124-125: Please provide further detail for readers on the Shapley decomposition method

• Lines 160-163; 165-168: Please clarify how the percentage of explained variance is obtained or calculated. Can this be discerned from Figure 2? In the figure 2 caption, please provide greater detail and clarify what the x-axis represents.

• Discussion: The last two paragraphs in the discussion section seem like they should be swapped. Ending the manuscript by discussing the limitations minimizes the impact of the work so I would suggest addressing limitations and then finishing with the concluding paragraph.

Minor Concerns:

• Line 51: “Higher-ranked”

• Line 85: Word missing? “…study (to) the…”

• Line 103 and throughout: I believe text concerning the methodology should be in past tense (i.e. “…we estimated logistic…)

• Line 123: Please briefly clarify for readers why probit and linear probability models were estimated in addition to a logit model.

• Line 133: Please write out what “pp” indicates before using the abbreviation

• Lines 170-172: Seems like a word or two is missing in this sentence as it is unclear what is being discussed (“…while the contribution of country-level preferences…”)

• Lines 186-187: Please discuss further the result that 15% of the withdrawal probability explained can be attributed to gender. This seems low, especially given the disparity in withdrawals between males and females. What could explain the other 85%? Is the low percentage explained a limitation and does it hinder the ability to draw strong conclusions about gender differences from these data?

• Do females typically have longer professional careers than males? If so, they may feel they are better able to accommodate skipping a tournament. Just a thought.

Reviewer #2: Referee report on:

“The Gender Gap in Aversion to COVID-19 Exposure: Evidence from Professional Tennis”

Manuscript: PONE-D-38751

PLOS One

Summary and General Assessment:

This paper studies gender differences in aversion to risks related to the COVID-19 Pandemic using the setting of the 2020 US Open. The authors find that female players are more likely to withdraw from the tournament compared to men, even after controlling for country-characteristics. About 15% of the withdrawal rate is explained by gender. .

In general, there are many things to like about this paper, the setting is novel and well suited to study this question, the paper is written in excellent English, and the analysis is competently executed.

Main Comments:

1. I would like to see a larger discussion about the weaknesses of the study, most centrally about the statistical inference given the small sample size. I know that this is mentioned briefly in the last paragraph, however the authors should think carefully about if there is evidence that would support these findings given a larger dataset. Additionally, if there is a larger sample which could be used as a robustness check, this would greatly strengthen the analysis.

2. The introduction and beginning of the paper should include a larger discussion about changes in risk taking throughout the COVID pandemic. A number of papers now examine how preferences and beliefs over risk change due to exposure to the COVID pandemic (e.g., Bu et al 2020). I think that this study needs to link to that larger literature and perhaps a brief discussion more fundamentally if the authors observations are driven by changing preferences/beliefs about risk due to exposure, or if their effects are more driven by innate gender differences towards risk.

3. The authors may also consider gender differences in expectations and beliefs in other domains, and how this may motivate their study, for example, D’Acunto et al (2019).

References

D’Acunto, F., Malmendier, U., Ospina, J. and Weber, M., 2019. Exposure to daily price changes and inflation expectations. Forthcoming Journal of Political Economy

Bu, Di and Hanspal, Tobin and Liao, Yin and Liu, Yong. 2020. Risk Taking, Preferences, and Beliefs: Evidence from Wuhan. Working Paper.

6. PLOS authors have the option to publish the peer review history of their article (what does this mean?). If published, this will include your full peer review and any attached files.

Reviewer #1: No

Reviewer #2: No

---

## [Author Response · Author response to Decision Letter 0]

23 Feb 2021

Response to editor:

Thank you for your consideration of our manuscript titled titled “The gender gap in aversion to COVID-19 exposure: evidence from professional tennis” for consideration by the PLOS ONE, and for the reviews of our paper.

We have revised the paper accordingly and we believe that the paper has been improved. In line with the editor and reviewers’ suggestions, we have edited the motivation and conclusions of the study to locate it in a broader context of gender differences in risk aversion and expectations, and of lessons from social and behavioural science for pandemic response. We have added a discussion of alternative explanations to which the reviewers hinted to, and weaknesses of our study. We have edited the entire draft to improve the clarity of presentation and readability, and elaborated the implications of our findings. We have also replaced Figure2, which was found confusing by one of the reviewers, by a table that conveys more information.

We hope that you would find the revised draft worth publishing in PLOS ONE.

Response to Reviewer 1:

Thank you for your comments and suggestions which we have accounted for in the revised draft. We have elaborated the implications of our findings in the “conclusions” section. We have also reorganized the manuscript and edited the presentation of our results and findings in order to facilitate readability. Below we respond to each of your main comments.

• Lines 50-56: As this paragraph involves discussion of the results, it seems it should be moved out of the introduction of the manuscript

We would prefer to retain this paragraph as it helps us to show our contribution.

• Lines 124-125: Please provide further detail for readers on the Shapley decomposition method

We have explained the method in more detail.

• Lines 160-163; 165-168: Please clarify how the percentage of explained variance is obtained or calculated. Can this be discerned from Figure 2? In the figure 2 caption, please provide greater detail and clarify what the x-axis represents.

We have replaced figure 2 with a table (Table 3) that includes the contributions of particular factors to overall variance, as well as their shares in total explained variance which were discussed in the text but not shown explicitly in figure 2. In the current draft, all data cited in the text are shown in Table 3.

• Discussion: The last two paragraphs in the discussion section seem like they should be swapped. Ending the manuscript by discussing the limitations minimizes the impact of the work so I would suggest addressing limitations and then finishing with the concluding paragraph.

We have swapped these paragraphs and elaborated the (now final) paragraph on broader implications of our findings.

• Line 51: “Higher-ranked”

• Line 85: Word missing? “…study (to) the…”

• Line 103 and throughout: I believe text concerning the methodology should be in past tense (i.e. “…we estimated logistic…)

• Line 133: Please write out what “pp” indicates before using the abbreviation

• Lines 170-172: Seems like a word or two is missing in this sentence as it is unclear what is being discussed (“…while the contribution of country-level preferences…”)

We have edited the text accordingly.

• Line 123: Please briefly clarify for readers why probit and linear probability models were estimated in addition to a logit model.

We have added an explanation: we do it in order to test whether our findings are robust to changes in estimation method. We find that our results do not depend on choice of the estimation method.

• Lines 186-187: Please discuss further the result that 15% of the withdrawal probability explained can be attributed to gender. This seems low, especially given the disparity in withdrawals between males and females. What could explain the other 85%? Is the low percentage explained a limitation and does it hinder the ability to draw strong conclusions about gender differences from these data?

We have elaborated this discussion. Overall, our models explain 20-30% of the overall variance in the total sample and various subsamples. We think it’s a decent share considering that the decision about participation may be affected by individual factors and past experiences, but the data allows to control only for sex, age, and ranking, and country-level variables (cultural preferences, level of development).

• Do females typically have longer professional careers than males? If so, they may feel they are better able to accommodate skipping a tournament. Just a thought.

In general, women reach their highest career level earlier than men, which is consistent with their more precocious biological development. Indeed, female players in our sample are on average younger than male players by about a year. However, the average career length of female players is not significantly different than that of male players (Guillaume et al., 2011). It is therefore unlikely that the differences in career life cycle can explain gender differences in the willingness to perform during the pandemic. We have added the discussion of this potential explanation to the “conclusions and discussion” section.

References:

Guillaume M, et al. Success and decline: top 10 tennis players follow a biphasic course. Med Sci Sports Exerc. 2011 Nov 43(11):2148–54.

Response to Reviewer 2:

Thank you for your comments and suggestions which we have accounted for in the revised draft. Below we respond to each of your main comments.

1. I would like to see a larger discussion about the weaknesses of the study, most centrally about the statistical inference given the small sample size. I know that this is mentioned briefly in the last paragraph, however the authors should think carefully about if there is evidence that would support these findings given a larger dataset. Additionally, if there is a larger sample which could be used as a robustness check, this would greatly strengthen the analysis.

We are aware that expanding our dataset with other tournaments would improve the analysis. However, the US Open was a unique natural experiment and there is no larger sample that can be used as a robustness check. The number of players eligible to participate was constrained. We are not aware of any other main, international sport competitions with identical rules and prize money for men and women and held in a country severely affected by the pandemic, such as the US. Therefore, it is not possible to find a larger data set that would support our hypothesis that the gender differences in the perception of COVID-19 were crucial in deciding whether to participate in the tournament.

We have expanded the relevant discussion in the conclusions section. We think that the differences in the number of COVID cases among top 100 male (10) and female (0) players is a suggestive evidence of the gender differences in risk aversion which we identify in our study. Also, studies of top professional tennis players tend to be based on relatively small samples (200-250 individuals, Guillaume et al., 2011), because of the nature of the sport.

2. The introduction and beginning of the paper should include a larger discussion about changes in risk taking throughout the COVID pandemic. A number of papers now examine how preferences and beliefs over risk change due to exposure to the COVID pandemic (e.g., Bu et al 2020). I think that this study needs to link to that larger literature and perhaps a brief discussion more fundamentally if the authors observations are driven by changing preferences/beliefs about risk due to exposure, or if their effects are more driven by innate gender differences towards risk.

We added a disclaimer that the decision about participation may be affected by individual factors and, most of all, personal experience, as noted by Bu et al (2020). However, our data does not allow controlling for personal exposure to the virus or any changes in preferences over time. Moreover, because the tournament that constitutes a natural experiment in our study was the first major tournament after the season was paused due to COVID-19 concerns, we cannot assess if the risk preferences of players have changed over the course of the pandemic. However, as our findings are consistent with previous evidence of higher risk aversion among women, we tend to interpret the effects we find as driven by innated gender differences towards risk, in particular to a specific type of risk related to pandemic exposure. Finally, because no outbreak occurred, we think that this experience is unlikely to have a long-term impact on risk aversion – because such effects are found for individuals affected by negative shocks and experiences (e.g. Malmendier and Nagel, 2011).

3. The authors may also consider gender differences in expectations and beliefs in other domains, and how this may motivate their study, for example, D’Acunto et al (2019).

Thank you for suggesting these studies. In the introduction, we expanded the discussion of research on gender differences in risk taking in various settings and contexts. We have also added more references, including the study by Bu et al. (2020), in the discussion of our results, which have expanded to discuss broader implications of our study. 

References

Bu, Di and Hanspal, Tobin and Liao, Yin and Liu, Yong. 2020. Risk Taking, Preferences, and Beliefs: Evidence from Wuhan. Working Paper. 

Guillaume M, et al. Success and decline: top 10 tennis players follow a biphasic course. Med Sci Sports Exerc. 2011 Nov 43(11):2148–54.

Malmendier U, Nagel S. Depression Babies: Do Macroeconomic Experiences Affect Risk Taking? The Quarterly Journal of Economics. 2011 Feb 1;126(1):373–416.

---

## [Decision Letter · Decision Letter 1]

10 Mar 2021

The gender gap in aversion to COVID-19 exposure: evidence from professional tennis

PONE-D-20-38751R1

Dear Dr. Lewandowski,

We’re pleased to inform you that your manuscript has been judged scientifically suitable for publication and will be formally accepted for publication once it meets all outstanding technical requirements.

Kind regards,

Valerio Capraro

Academic Editor

PLOS ONE

Additional Editor Comments (optional):

Reviewers' comments:

Reviewer's Responses to Questions

**Comments to the Author**

1. If the authors have adequately addressed your comments raised in a previous round of review and you feel that this manuscript is now acceptable for publication, you may indicate that here to bypass the “Comments to the Author” section, enter your conflict of interest statement in the “Confidential to Editor” section, and submit your "Accept" recommendation.

Reviewer #1: All comments have been addressed

Reviewer #2: All comments have been addressed

2. Is the manuscript technically sound, and do the data support the conclusions?

Reviewer #1: Yes

Reviewer #2: Yes

3. Has the statistical analysis been performed appropriately and rigorously? 

Reviewer #1: Yes

Reviewer #2: Yes

4. Have the authors made all data underlying the findings in their manuscript fully available?

Reviewer #1: Yes

Reviewer #2: Yes

5. Is the manuscript presented in an intelligible fashion and written in standard English?

Reviewer #1: Yes

Reviewer #2: Yes

6. Review Comments to the Author

Reviewer #1: The authors did a nice job incorporating suggestions that has lead to a greatly improved manuscript in terms of the implications and context of the results and overall readability. While I still feel like the introduction paragraph that discusses results (lines 55-61) belongs later in the manuscript, I can appreciate the authors rationale for including earlier and do not feel like it detracts from the paper. Thus, I am recommending acceptance of this manuscript.

Reviewer #2: I think the authors did an excellent job incorporating the feedback from the reviewers into the revised manuscript.

7. PLOS authors have the option to publish the peer review history of their article (what does this mean?). If published, this will include your full peer review and any attached files.

Reviewer #1: No

Reviewer #2: No

---

## [Editor Report · Acceptance letter]

12 Mar 2021

PONE-D-20-38751R1 

The gender gap in aversion to COVID-19 exposure: evidence from professional tennis 

Dear Dr. Lewandowski:

I'm pleased to inform you that your manuscript has been deemed suitable for publication in PLOS ONE. Congratulations! Your manuscript is now with our production department. 

Kind regards, 

on behalf of

Dr. Valerio Capraro 

Academic Editor

PLOS ONE